# Development of a service blueprint for blockchain services

Hyeji Jang[1], Do-Hyeon Ryu [2,3]*

1 Department of Industrial and Management Engineering, Pohang University of Science and Technology (POSTECH), Pohang, South Korea, 2 Department of Industrial and Management Engineering, Incheon National University (INU), Incheon, South Korea, 3 Research Institute for Engineering and Technology, Incheon National University (INU), Incheon, South Korea

* dhryu@inu.ac.kr

**Data Availability Statement:** This study is based on literature review, qualitative analysis (including open card sorting and comparisons with existing service blueprints), and expert discussion and evaluation, without the collection of primary data. As a result, no data was generated or collected, and there is no data available to share.

## Abstract

As blockchain has been actively applied in various services, a tool for visualizing the complex service processes reflecting the characteristics of blockchain has been required. A service blueprint is a tool to visualize all key systems and encounters in service delivery. Although several blueprints already exist, they have limitations to systematically visualize and analyze blockchain service processes. This study develops a Blockchain Service Blueprint (BSB) specialized in visualizing and analyzing blockchain service processes. A comprehensive literature review and an analysis of blockchain services were conducted to identify characteristics of blockchain services and limitations of existing blueprints. The BSB was developed based on the derived key components of blockchain service processes, so that it has the optimal structure with key elements to visualize complex processes of blockchain services. The usefulness of the BSB was verified by both comparisons with traditional blueprints and expert interviews. The proposed BSB can intuitively and clearly visualize a service process between customers and service providers in blockchain services. Using the Blockchain Service Blueprint (BSB), providers can identify and improve service processes to enhance sustainability, and this study offers researchers cases and a development process that demonstrate the BSB's effectiveness across various blockchain services.

## Introduction

Recently, the massive amount of international trade happens in service sector, such as financial and information services. In the sector, multiple companies offer similar services, and their business could be threatened [1, 2]. To overcome the threat, service companies have been trying to increase their competitiveness by effectively managing business processes which includes activities and workflows in services to create values [3]. In addition, service companies can refine their service processes and provide customers better experiences by adopting new technologies such as artificial intelligence and blockchain technologies. The efforts that consider better service processes and new technology adoption are essential tasks for service companies [4, 5]. This study deals with a tool to visualize and analyze service processes that adopt new technology called blockchain.

**Funding:** This work was supported by the IITP (Institute of Information & Communications Technology Planning & Evaluation)-ICAN(ICT Challenge and Advanced Network of HRD) grant funded by the Korean government (Ministry of Science and ICT)(IITP-2025-RS-2023-00260175) awarded to DHR.

**Competing interests:** The authors have declared that no competing interests exist.

**Abbreviations: BP**, Brain Power; **BSB**, Blockchain Service Blueprint; **CSB**, Conventional service blueprint; **DApp**, Decentralized application; **ISB**, Information Service Blueprint; **MED**, Medi Token; **MP**, Medi Point; **O2O SB**, Service blueprint for the online-to-offline integration; **O2O**, Online-to-offline; **P2P**, Peer-to-Peer; **PoS**, Proof of stake; **SEB**, Service Experience Blueprint.

Blockchain is a type of distributed system managed through a peer-to-peer network. Blockchain stores and manages the ledger containing transaction information not through a central server, but multiple participants connected to the blockchain network. In this time, data is recorded in a connected block structure like a chain [6]. Blockchain has the advantage of ensuring high security, ensuring speed, and transparency of the transaction process by applying distributed data processing and encryption technology at the same time. The way data is recorded and managed on the blockchain can effectively defend against external attacks and data distortion [7].

Blockchain has had a great influence on the global service industry with its totally new paradigm of dealing data [8–10]. The use of blockchain enables transactions without a third party, reducing transaction costs [11, 12]. In addition, using cryptocurrency that has no borders reduces the transaction time across the countries. In addition, the encrypted and distributed data management method increases the transparency of the data and ensures excellent anonymity for network participants [13]. As a result, services that apply blockchain can easily be operated in a healthy way, provide fair compensation, and detect malicious users without arbitrary censorship of operating institutions [14, 15]. Above these, the various characteristics of blockchain can motivate service users to actively use and maintain the high quality of the service. Recently, a new type of service that actively utilizes blockchain, called blockchain services, is being launched. Especially in multinational enterprises, efforts to introduce blockchain are continuing in many existing global services to take its advantages, such as improving a process of cross-border payment [12, 16, 17].

Despite the numerous advantages of blockchain, not all blockchain services succeed. One of the main reasons for service failure is that service providers and developers do not fully understand processes of their services [18]. Such lack of understanding could cause problems. For instance, they do not know where the important touchpoint in which various interactions between customers and service providers happen. Therefore, they cannot improve their processes effectively and prevent serious problems that cause negative customer experience. For this reason, understanding service processes is important for the success of their services [19]. The lack of understanding could also happen for blockchain services [20]. Service processes and corresponding activities in blockchain services are different from those of traditional services. For example, in traditional cross-border financial transactions, there are costs, such as currency exchange and fees, to transaction guarantors in each country. It is a very time-consuming process, but it is necessary for a secure transaction. In cross-border financial transactions that apply blockchain, however, users conduct transactions without third-party guarantees, shortening transaction times and creating a new fee policy [21]. As the number of blockchain services increases globally, blockchain creates global issues, such as forcing distributed users around the world to face a new environment without strong central control and to use virtual currencies without borders [22]. For this reason, service providers and developers have devised a method to understand processes of blockchain services, and such efforts are becoming important globally for the success of blockchain services. Accordingly, understanding the changed process and user interaction of blockchain services is emerging as a key problem [23, 24].

International business experienced the way of transition from a product to a service [25]. During the transition, various service design methods have been developed for the business [26]. A service blueprint is one of the methods to design and develop a service process [27]. It visualizes a service-delivery process and the relationships among various service components from the customer's viewpoint to help service providers and developers understand the service processes [28, 29]. A blueprint can be a good tool to understand processes of blockchain services, so that mangers can identify their problems and improvement points for their success.

Although several blueprints, such as conventional service blueprint (CSB) [30], information service blueprint (ISB) [31], service experience blueprint (SEB) [32], and O2O SB [33], exist, such blueprints have limitations to cover the functions, items, and processes of blockchain services. A new service-blueprinting framework for a specific service has been developed when extending and adapting the existing tools cannot successfully present new service processes [31, 32, 34]. Therefore, this study develops a new blueprint specialized in blockchain services to help service providers and developers understand effectively processes of their services.

Therefore, the main aims of this study were to:

1. Understand the characteristics of blockchain services;

2. Develop and verify a new service blueprint that can effectively visualize processes of block-chain services;

3. Provide managerial implications of the developed service blueprint for blockchain service providers.

The remainder of this study is as follows. The literature review introduces blockchain and blockchain services and describes existing service blueprints and their limitations. The method section details the development of the blockchain service blueprint (BSB). The next section outlines essential components of blockchain services, highlighting key characteristics. The blockchain service blueprint section presents the BSB and its illustrations with two blockchain services, and the result of expert interviews. The discussion compares the BSB with the three existing service blueprints, namely, CSB [30], ISB [31], and SEB [32] and presents the managerial implications. Finally, the study concludes with the contributions and future research directions.

## Literature review

This section explains the unique characteristics of blockchain and blockchain services in detail, then presents existing service blueprints and their limitations in visualizing processes of block-chain services.

### Blockchain characteristics in services

Blockchain is a distributed ledger that records transactions, maintained independently by each participant to ensure data consistency and security [17]. A set of data that should be managed is called a block, and blocks are connected in a chain form by a consensus algorithm, allowing unified decision-making without central authority [35]. Since all participants have the same copy of the data, no specific participant can arbitrarily modify them [36]. All users in the blockchain network can store the data, so the system remains stable even if some of them are disconnected. Therefore, it is tolerant to fault caused by unstable or even unreliable participants [35, 37].

A defining feature of blockchain is the smart contract, which allows direct and automated transactions without intermediaries [8]. Smart contracts, implemented as code, execute when pre-set conditions are met, ensuring transactions are secure, trackable, and irreversible [38, 39]. Introduced by [40] and expanded with blockchain technology, smart contracts are widely used in applications requiring verifiable self-executing agreements, such as real estate, identification, and supply chain services [41, 42]. For example, in blockchain-based real estate, a smart contract replaces brokers, verifying ownership and payment through decentralized consensus. Therefore, it reduces costs and mitigating fraud risks [43, 44]. Because of this

advantage, smart contracts have been actively utilized in various blockchain service industries, such as identification, supply chain, energy, and IoT.

A combination of service processes with tokens/coins is a significant evolution in blockchain services, providing exchangeable value as a monetary means [45, 46]. The biggest difference between tokens and coins lies in platform independence; coins operate on independent blockchains such as Bitcoin, while tokens function within other platforms [47]. Coins generally serve as native currencies within specific systems, while tokens offer utility functions, such as facilitating community interactions [48, 49]. Tokenization encapsulates diverse types of value, allowing them to be represented and managed across distributed networks [50–52]. This concept supports the representation of real-world asset values and trust within the network, thus expanding the potential applications of blockchain to manage various assets effectively [45, 53].

Tokens and coins also serve as incentives to promote specific activities within blockchains [54, 55]. For example, blockchain systems reward users who maintain the system through mining by providing coins, a method that contrasts with traditional centralized management. On social media platforms like Steemit, tokens as rewarded to users who create or curate high-quality content, democratizing revenue opportunities that are typically limited to influencers in conventional platforms [45]. This incentivized reward system fosters sustainable engagement, encouraging positive contributions from users without centralized oversight. Such mechanisms introduce new users, enhance desire behaviors, and maintain overall system health, supporting a self-sustaining blockchain service ecosystem [56, 57].

Due to these advantages, blockchain has been actively applied to services in various fields. In finance, it reduces transaction costs by simplifying processes, while in social media, such as Steemit or Everipedia, it fosters decentralized moderation and content value assignment [58]. Gaming services leverage blockchain to attribute real-world value to virtual assets, thus motivating user participation. In healthcare, blockchain offers users direct control over personal data, improving both security and data-sharing convenience. Blockchain's applicability in these sectors showcases its potential for secure and efficient service delivery, supporting users' control over their own information, transactions, and content contributions across [17, 43].

Recently, blockchain has expanded into international business, where multinational companies integrate it for operational efficiencies [12, 59]. However, the focus in research often remains on blockchain's benefits in marketing, advertising, banking, overlooking the complexities involved in implementation. Blockchain introduces fundamentally different service processes, affecting system configurations, service operations, and customer journeys [60, 61]. To fully leverage blockchain's benefits, businesses have to approach integration thoughtfully, recognizing the profound shifts it brings to service processes. Tools like service blueprints can assist in visualizing these changes, helping developers and operators design blockchain-based services effectively by mapping unique, decentralized service workflows [31, 62].

### Existing service blueprints and their limitations

A service blueprint (SB) is a design tool to map out dynamic interactions, key touchpoints, and potential bottlenecks in service processes [30]. By highlighting the customer journey, a SB can identify what enhances or impedes customer experience [32]. Typically, the vertical axis of a blueprint divides components, such as physical evidence, customer actions, and support systems, to visualize interactions between customers and providers. Service activities are chronologically arranged on the horizontal axis, aiding service providers in designing effective customer interactions [33]. Because of these advantages, various SBs have been developed and

applied across various services, including hotel services [63], smart product services [64], and healthcare services [65].

A comprehensive review was conducted to evaluate SB frameworks for visualizing blockchain services. Various SBs were identified on Google Scholar by using keywords such as "blueprint," "service blueprints," "service process design," and "service process management." Among the SBs, specific SBs that are representative and focus on a process of information with various touchpoints between providers and customers were selected. From this review, three prominent SBs, the customer service blueprint (CSB) [30, 66], the service experience blueprint (SEB) [32], and the information service blueprint (ISB) [31], were selected. These SBs include frameworks hat emphasize the customer's journey and key interactions with service components, making them relevant for detailed consideration of blockchain's unique requirements. Despite the variety of existing SBs, the need for specific adaptations to suit blockchain's decentralized and token-driven nature is evident, as traditional SBs may not fully capture the nuances of blockchain service flows.

The CSB [30, 66] emphasizes customers as central in service processes, a characteristic beneficial for understanding customer requirements and improving service offerings [30],. The CSB's structure makes it. N adaptable reference for other SBs and serve as a useful framework for blockchain services, which include core service components like customers, service providers, and systems. The SEB [32] was developed to visualize multiple interfaces as touchpoints between users and service developers. It allows a detailed view of user interactions with both people and machines like tokens and coins. This makes SEB particularly relevant for blockchain services, where decentralized digital interactions and user engagement with digital assets are frequent and complex.

With the rise of online information exchanges, the ISB [31] focuses on visualizing information-intensive service processes, highlighting interactions between service providers and customers. Similarly, the online-to-offline (O2O) SB [33], which facilitates O2O integration, helps in understanding integrated service processes between online and offline as a single service-delivery process. These SBs enable visualization of diverse, multi-channel service flows. However, despite their strengths, existing SBs lack dedicated components to capture blockchain's decentralized features and processes, particularly around tokens, coins, and transactions. As a result, adapting these SBs for blockchain services often involves combining multiple blueprints, which can complicate understanding the entire service flow and identifying key touchpoints.

While existing SBs can be modified to present new service processes. However, if changing foundational components and structure is required, a new service blueprinting framework for a specific service has been developed [31, 32, 34]. The limitations explained above imply that new components and structure are necessary to visualize unique processes of blockchain services. Development processes of existing blueprints include finding key components that are essential for a structure to effectively visualize and analyze service processes and validating whether developed blueprints can address the limits of existing SBs. With the developed SB in this study, a user can present processes of blockchain services without extra effort and time in drawing and combining multiple SBs.

## Method

This study was composed of 4 steps to develop the BSB (Fig 1). First, literature and blockchain services in various domains were reviewed to extract representative keywords used to explain processes of blockchain services (Fig 2). Relevant literature was collected from Google Scholar to identify the impact of blockchain on services. Google Scholar covers more journals and

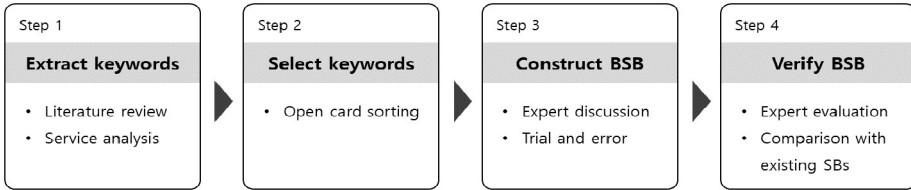

**Fig 1. Overview of the development process of the BSB.**

research areas than other research databases, such as PubMed, Scopus, and Web of Science [67]. By using keywords, such as blockchain service, blockchain application, and potential of blockchain, literature after 2008, when blockchain became known to the public, was collected. Only peer-reviewed journals were included to ensure content reliability. Literature not focused on service processes or user activities, such as those on improving blockchain consensus algorithm, was excluded. As a result, 39 papers that provide cases or detailed explanations about service processes or user activities of blockchain services were selected and scrutinized.

In addition, blockchain services in operation, regardless of service field, were collected. Then, only services that provide white paper and are mature enough to analyze service processes were selected. In addition, to accurately grasp the changes due to the adoption of blockchain, only the services in which blockchain is the core technology were selected rather than existing services with miner introduction of blockchain. Experts conducted task analysis on 89 selected blockchain services, referencing white papers. Task analysis systematically examines the interaction between humans and services and service flow. By recording the purpose, initiation point, and action of the performed task, the service process and problems can efficiently be identified [68]. After performing both literature review and service analysis, three experts

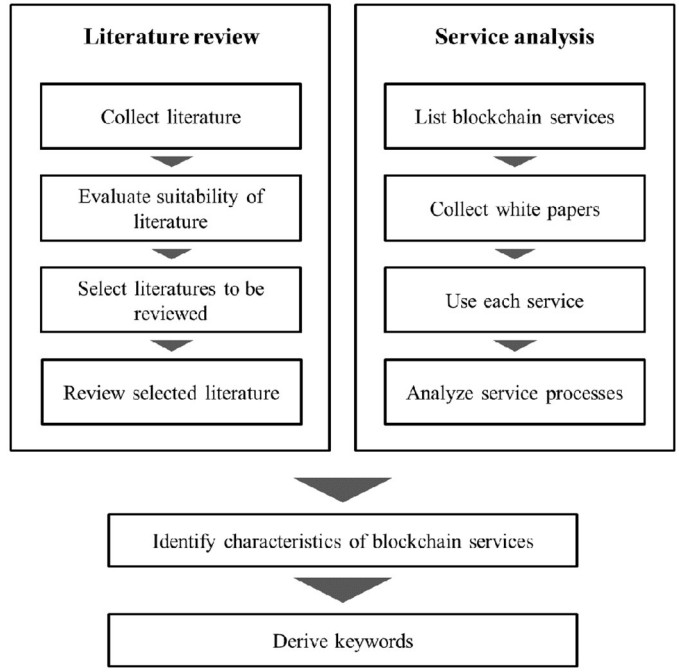

**Fig 2. Flows of literature review and service analysis in step 1.**

who were searched and accepted our interviews participated in a session to confirm and organize the acquired knowledge. The experts have sufficient knowledge of the principles and applications of blockchain and experience in conducting research on the design and development of blockchain services in academia. Their average career is 4.6 years. In this session, they identified how blockchain impacts service processes and the difficulties users face. Different points from the existing service processes were figured out as a keyword after consensus was reached.

Extracted keywords were refined in the second step, open card sorting. Open card sorting is a general method to derive specified information from participants [69]. It is used to derive upper categories by collecting similar cards in a bottom-up manner without a pre-set category. Repeated keywords were combined, and keywords that have the same meaning were standardized and unified, then all keywords were organized using open card sorting. As a result, four keywords that must be considered as features when expressing the process of blockchain services were derived: interaction between users, tokens and coins, trading time, and operation platform. These features were used as the foundation of the BSB.

The BSB structure was constructed to fulfill the features identified in the second step. A blueprint should have a structure that can visualize complex service processes as intuitive and clear processes, so various alternative structures were compared by mapping important processes of blockchain services in various fields. 89 services were utilized in this step. Three researchers made repeated modifications by mapping various user activities that occur in the blockchain service and discussing the results. Trial-and-error eventually yielded a structure that appeared optimal.

Finally, the developed BSB was verified using two methods. First, five experts in blockchain services evaluated the usefulness of the BSB. They were shown the essential components of the BSB and some visualized examples, then they evaluated the BSB according to four criteria: completeness, effectiveness, efficiency, and versatility. In addition, qualitative opinions about the strengths of the BSB and potential for improvement were collected using interviews. Second, the utility of the BSB was evaluated by comparing it to traditional SBs, i.e., the CSB or ISB with the same services. These tests determined that the BSB can overcome the limitations encountered when mapping blockchain services to traditional SBs.

## Essential components of blockchain services

Literature related to blockchain technology and blockchain services was reviewed to identify key features of blockchain services that differ from features of traditional services. Each feature was categorized using open card sorting; the result detected four features (Table 1) that must be considered when expressing the process of blockchain service: Interaction between users,

**Table 1. Findings from literature review.**

| Feature | Related keywords | Related studies | References |
|---|---|---|---|
| Interaction between users | Distributed system, no third party, smart contract | Distributed storage of data using blockchain, self-cleaning of the service through the communication between users, direct transactions between users without intermediaries with active use of smart contracts | [15, 44, 50, 70–78] |
| Tokens and coins | Token economy, reward | Maintaining the system using coins as a reward for mining, using tokens to induce desirable behavior of users | [46, 57, 79–84] |
| Trading item | Tokenization, P2P trade | Easy to trade by tokenizing physical and non-physical assets, objectively split ownership of physical assets, secure P2P transactions for various assets | [47, 48, 50–53, 85–89] |
| Operation platform | Service provider, recentralization | Manage the service platform to operate stably, stablish service rules | [90–95] |

tokens and coins, trading item, and operation platform. These four features are the major components of the BSB. The rest of this section explains each component in detail.

The most prominent feature is the importance of interaction between users. In blockchain services, the term 'user' encompasses a broader range of roles than in traditional services. Various stakeholders such as miners and verifiers are included in the user, and the interaction between them plays an important role in the stable operation of the blockchain service. In blockchain services, two factors can promote user interactions. First, blockchain is basically a pure distributed system [50, 70]. Blockchain leverages distributed ledgers and technologies to decentralize authority. Users of blockchain services actively interact with others and perform roles traditionally managed by a central institution. For example, introducing blockchain to a news platform can keep services healthy by mutual censorship among users. Smart contracts facilitate user interactions by enabling user transactions without intermediaries. Users can directly exchange items such as money and information. Because of this characteristic, interactions between users are essential in blockchain services.

Tokens and coins are crucial features of blockchain services [83]. They can be used for various purposes in blockchain services [80, 96]. Tokens and coins are used to store and convey monetary value [57, 81, 82], or to represent ownership of certain assets [54, 55, 96]. Proper use of tokens and coins helps to attract users to the service and keeps the service healthy. Service developer must select appropriate tokens or coins to support processes. The exchange process of tokens for coins should also be carefully designed. Clear and accessible token-related information is essential for users.

In blockchain services, various items can be traded. Similar to traditional services, transaction information such as login information and contents is created. The feature of blockchain that is most distinct from existing services is that it supports control of ownership of both non-physical and physical attributes by a process called tokenization. By using tokens during the data-transfer or data-storage stages, the original data can be reliably protected. Tokenization allows easy ownership claims and asset divisibility [48, 52]. Trading of patents, intangible assets such as copyrights, alternative assets such as gold and iron, and irreplaceable assets are all possible through tokenization. Integrating tokenization and smart contracts enables safe and efficient transactions without third parties. P2P transactions that use blockchain are being actively researched, especially in the energy field [85, 87, 89].

The central authority is noticeably weakened in blockchain services, but the role of the service operator remains [90–92]. An operation platform functions similarly to traditional support systems in maintaining services [30, 33, 93, 94]. Users can interact with the operation platform through channels such as websites and DApps. However, the blockchain service operator performs the unique function of distributing tokens to users according to activities, recording smart contracts, and establishing rules to discover malevolent users on operation platform [91].

The characteristics of blockchain services significantly impact user experience. To maximize the positive effect, blockchain services should be carefully designed considering the changes that blockchain brings. To successfully design a blockchain service, service developers must clearly recognize the changes in service processes brought by the blockchain and understand activities of users accordingly. Tools for analyzing service processes and user activities are therefore essential [97].

## Blockchain service blueprint

This section first explains the components of the BSB and its structure. The components were extracted based on the characteristics of blockchain services introduced in the previous

section. The structure was developed through iterative mapping of various blockchain services. Second, two cases were illustrated with the BSB. The validation assessed whether the BSB includes essential components for visualizing service delivery processes and an adaptable structure for mapping diverse services that have different characteristics. This method was adopted in previous studies to validate blueprints [30–32]. Finally, expert interviews were conducted to validate the usefulness of the BSB. Five experts evaluated four criteria: completeness, effectiveness, efficiency, and versatility.

## Structure of the blockchain service blueprint

The BSB (Fig 3) consists of four unique elements: User Activities, Tokens and coins, Transaction Items, and Operation Platform. User activities, tokens and coins, and transaction items should be visualized for each user. The three rows are arranged both ways along the side of the Operation Platform row. This structure intuitively visualizes the service process from the user's perspective. For example, the structure shows user approach, goals, and service process user experienced. The structure can also show which processes of different users are connected to meet their requirements. Service developers can quickly identify touch points at a glance and the exchange of tokens, coins, and transaction items. Placing the Operation Platform centrally enables service developers to examine if platform actions meet user requirements.

User activity refers to actions taken by users to achieve their objectives in a service. In blockchain, a P2P network is implemented in which each user can directly connect to, and trade with, other users. Unlike many traditional services, particularly in finance, which target specific regions, blockchain services are accessible to users globally. Users who interact with each other may be located in distant regions. Interactions such as registering, posting, and voting occur both between users and between users and the operation platform. Such activities should be visualized to understand why and how the interactions occur. Compared to the existing SBs that focus on the interaction of a single user with the service platform, the BSB can better visualize the interactions between users that occur frequently in a decentralized environment.

Tokens and coins are also important components that reflect the characteristics of blockchain services. Tokens and coins are digital assets, so they do not have physical limitation.

| | |
|---|---|
| **User 1 Activity** | Activities performed by service user 1 |
| **Token & Coin** | Tokens and coins related to the activities of user 1 |
| **Transaction Item** | Transaction items related to the activities of user 1 |
| **Operation Platform** | Roles of various stakeholders involved in service process |
| **Transaction Item** | Transaction items related to the activities of user 2 |
| **Token & Coin** | Tokens and coins related to the activities of user 2 |
| **User 2 Activity** | Activities performed by service user 2 |

**Fig 3. Structure of the BSB.**

Tokens and coins facilitate user activity and asset digitization in blockchain services. Therefore, it is crucial to visualize and understand the purpose and exchange of tokens and coins. Since various types of tokens and coins can exist in a blockchain service, they should be carefully presented.

Transaction items represent the outcomes of users' activities in a service. Users can exchange transaction items during the service processes. Transaction items range from non-physical items like personal private information to physical items like cars or houses. If transaction items are not shown in a service process, then users and service developers have difficulty understanding the interaction results generated among users or between users and a platform. Therefore, the transaction items used for each user's action must be visualized.

An operation platform, which may be a person, computer, or system, processes users' requirements. The operation platform manages multiple tasks, such as issuing tokens and managing smart contracts, maintaining a close relationship with users. When the requirements come through users, the platform takes actions to meet the requirements. For example, the operation platform should respond to the requirements quickly and accurately to ensure a high quality of service. To ensure effective responses, service developers should understand and analyze the functions and interactions of operation platforms within the BSB.

## Illustration of the blockchain service blueprint

To validate the generality and functionality of the BSB, many types of blockchain service were mapped, and two of them are presented in this paper. Generality refers to the BSB's coverage of diverse blockchain services, and functionality pertains to its capability to visualize and analyze service processes of a blockchain. Medibloc is a service that provides a platform to record, manage, and trade medical information of users. Various medical information-related transaction items are exchanged in the service. Everipedia is a service by which users share their knowledge. It ensures the high quality of content by providing users with the right to vote for the content. This service effectively illustrates user interactions and token utilization.

Medibloc is a service that provides diverse functions, such as managing and trading the medical records of users. The service includes two user types: general users and medical teams. A general user refers to a person who receives medical treatment and manages his or her medical record on the platform. A user in a medical team includes medical officials and researchers. Medical officials provide medical treatment to general users, and medical researchers conduct research with the medical records of users. MED and MP are used when interaction occurs, such as when a medical team purchases a general user's medical record or provides additional notes.

Fig 4 illustrates Medibloc service on the BSB. Users in a medical team are required to acquire the approval of other users who have already registered as medical teams. After registering and obtaining a private key, general users can enter their medical record on the platform. When users upload their medical record, the information is encoded using their private key. The user can obtain an MP as a reward for providing his medical record. When a general user receives medical treatment from the medical team, the user can authorize the medical team to access their medical record by providing advance consent. When the medical team enters the information on the service, MPs are given to the medical team and general user. Insurance claims automatically start by executing a smart contract. A medical team can earn MP by providing online diagnoses in response to general user inquires.

Fig 4 shows how interactions between users take place in a blockchain service. When user 1 receives treatment information from user 2, the two users must be visualized while they are using medical records recorded on the blockchain, and while they are paying medical fees to a

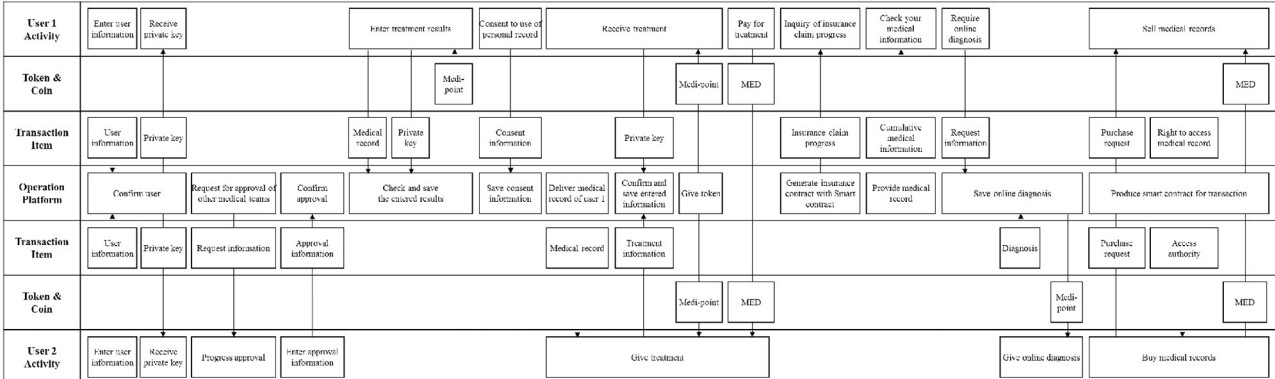

**Fig 4. Illustration of the Medibloc service on the BSB.**

MED. If only one user is presented in the BSB, then the entity that pays or receives MED is not easily identified. The Token & Coin row clearly depicts the direction and timing of MP and MED movements within the service process. Also, it can describe new items such as private keys that were not seen in the existing service. Lastly, the BSB can visualize the functions of the operation platform, such as creating a smart contract, confirming users, and saving information.

In Everipedia, users can suggest new knowledge and edit existing knowledge that other users wrote. The change history of contents is open to all users. The blockchain's consensus algorithm used by Everipedia is PoS, which means that users can mine or verify blocks in proportion to the number of coins that they own. In Everipedia, IQ coin and BP tokens are used. Users can obtain or use IQ and BP by voting to express their opinions about contents and by producing new contents. This system helps maintain high service quality.

The BSB can also illustrate the service process of Everipedia (Fig 5). In the illustration, two users register by obtaining and entering a verification code generated by the operation platform. Each user buys IQ, which are issued at regular intervals of time on the operation platform, to perform activities on the operation platform and stakes a certain number of IQs. IQ

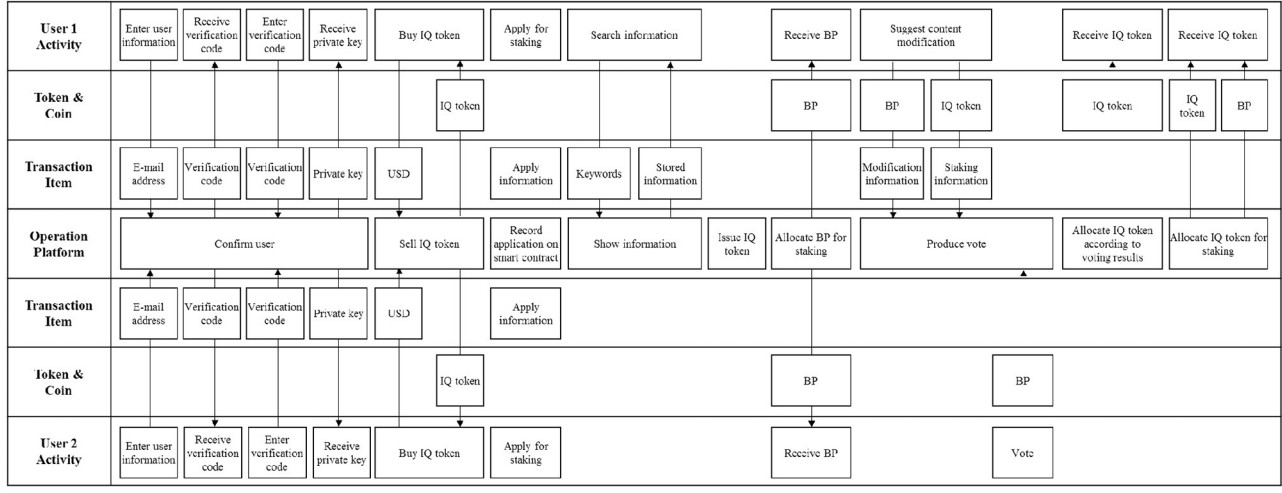

**Fig 5. Illustration of the Everipedia service on the BSB.**

staked by the smart contract can be used after a certain period of time, and the user gains BP as a reward for contribution to PoS in proportion to stakes. The users who have the BP can suggest editing the existing content or posting new content. At this time, the users should stake a certain amount of IQ and consume the BP so that the service prevents imprudent suggestions. When voting for the suggested content, the users who suggest content can obtain IQs as a reward depending upon the result of voting.

The BSB can effectively represent the case in which tokens and coins and other items move simultaneously (Fig 5). In blockchain services, transaction items and tokens and coins are often combined. For example, when a user modifies information, BP must be paid together, and for staking activity, IQ must be moved. When tokens and coins have different functions in the service, the BSB clearly visualizes this phenomenon by expressing them as separate rows. In the example of Everipedia, the new function of the operation platform is more prominent than the example of Medibloc. Activities like issuing and selling IQ, and staking BP, are unique to blockchain services.

These illustrations highlight the following points. First, the BSB can visualize the essential components that should be thoroughly considered when blockchain services are operated. In such services, understanding interactions among users, and between users and an operation platform, is important because key activities happen with coins and tokens during the interactions. The illustrations also show that complex processes were intuitively presented.

The symmetric structure of the BSB, with User Activity rows at the top and bottom, effectively visualizes individual user activities and interactions between users. The coins and tokens that are important elements in blockchain services can be also visualized in the BSB. Service developers should be able to understand the process of tokens and coins used for different purposes in service processes. The Token & coin rows assigned to each user can help service developers easily identify relevant tokens and coins for specific users and activities. Finally, the Transaction Item rows are also useful for service developers. The rows present the exchange of various items between users, either directly or through the operation platform. The BSB enables to avoid the confusion by separately visualizing tokens, coins, and transaction items in the distinct rows.

In this section, Medibloc and Everipedia were mapped on the BSB. These two services have different characteristics. In Medibloc, the two user groups have different functions: general users record and manage their medical records, and the medical team provides medical treatment and diagnosis. In contrast, in Everipedia, all users can have equal authority and perform similar activities during the service. All users can suggest and edit content, and vote on others' suggestions.

The types of tokens, coins, and transaction items vary across services. In Medibloc, MP and MED are used and various transaction items such as medical records, insurance claims, and online diagnosis are exchanged between users. However, in Everipedia, the BP and IQ are used between users, and they share one type of transaction item, namely, contents.

Finally, the complexity of functions between the two services is different. Medibloc has more complex functions than Everipedia. This demonstrates that the BSB can visualize diverse and complex service processes, regardless of service type. The illustrations validated that the BSB satisfies the need for generality and functionality as a tool to visualize different and complex processes of blockchain services.

## Evaluation of the blockchain service blueprint through expert interview

An experiment was conducted to validate the usefulness of the proposed BSB. Five experts with deep knowledge of blockchain technologies and features of blockchain services evaluated

**Table 2. Evaluation results of the BSB.**

| Criteria | Description | Mean (S.D) |
|---|---|---|
| Completeness | BSB has all necessary elements and structures to visualize the process of blockchain services | 4.8 (0.4) |
| Effectiveness | BSB enables service designers, developers, and providers to intuitively and accurately visualize the processes of blockchain services | 5.6 (1.1) |
| Efficiency | BSB enables service designers, developers, and providers to quickly and easily visualize processes of blockchain services | 5.6 (0.8) |
| Versatility | BSB can be used regardless of the field and types of blockchain service | 5.0 (0.7) |

whether the BSB is helpful to visualize service processes of blockchain services. The experts, selected from both academia and industry, specialized in blockchain services and have experience in developing business models and designing various blockchain services (e.g., medical care, travel, and knowledge sharing). Their average career is 3.6 years.

One-on-one interviews, each lasting approximately 1.5 hours, were conducted to gather expert feedback. The interview comprised three sessions: a description of the BSB, a quantitative evaluation, and an interview. An initial provided a brief explanation of the BSB's composition and application. Following a full review of the BSB, experts rated each evaluation criterion on a seven-point Likert scale (1: strongly disagree, 7: strongly agree). To evaluate the usefulness of the BSB, four criteria were selected: completeness, effectiveness, efficiency, and versatility (Table 2). All four criteria averaged above 4, indicating a positive effect of the BSB on the visualization of blockchain services.

Qualitative results captured expert opinions on the BSB's advantages from a service developer's perspective. From the viewpoint of service developers, all experts agreed that the BSB helps to view the overall processes at a glance compared to other service blueprints. They noted the BSB's efficiency in intuitively presenting key elements and interactions essential to service development. Additionally, they were satisfied that the structure of the BSB is specialized in visualizing P2P interactions that frequently happen in blockchain services, as well as the interactions that involve only one user. They expected that the BSB would be useful to service designers and providers as well as to developers, serving as a tool for cross-functional team communication. Finally, the experts were satisfied that the BSB has generality as a tool. Without generality, the BSB's elements and structure would require frequent adjustments, incurring significant time and effort.

## Discussion

In this section, we compare the BSB with the three existing service blueprints: the CSB, ISB, and SEB. The result shows that the existing blueprints have critical limitations in visualizing processes of blockchain services, which the proposed BSB completely addresses. Second, the managerial implications of the BSB, identified through literature review and service mapping, are discussed.

### Comparison of the BSB and existing service blueprints

The CSB focuses on visualizing service processes offline. It is the foundation of blueprinting framework research and is widely regarded as a benchmark. The ISB specializes in visualizing service processes online, especially the process of information. The SEB focuses on the actions of multiple participants and systems. These three blueprints were selected because blockchain service users engage in active interactions and generate significant information. Mediblock,

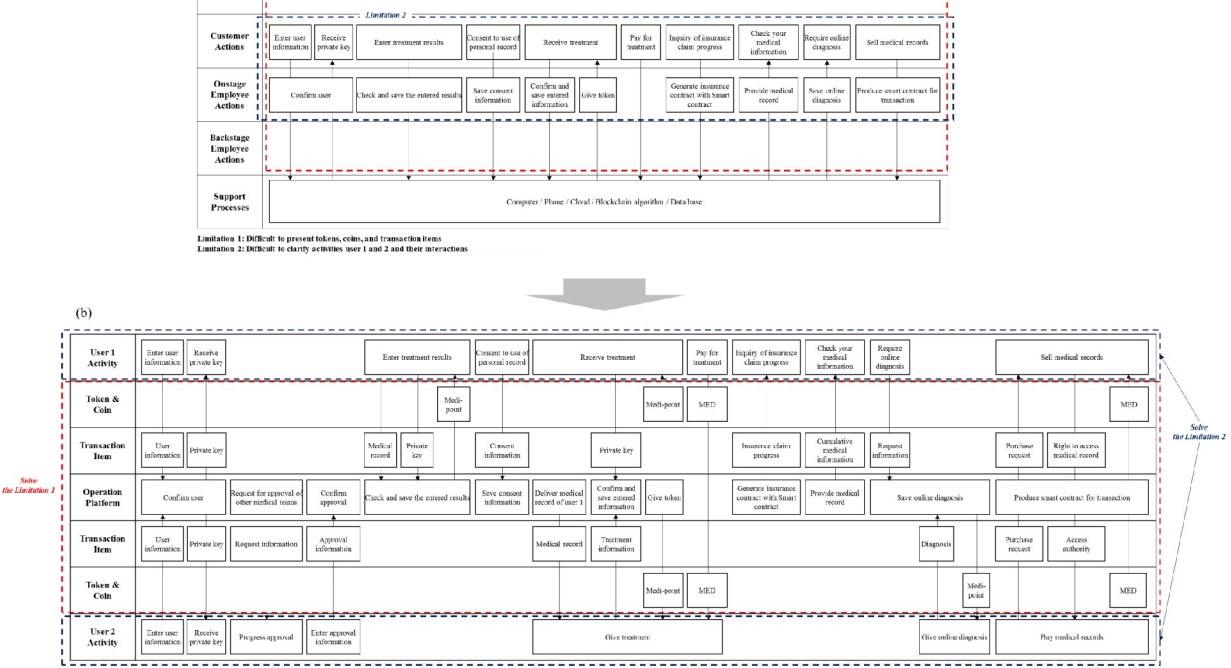

**Fig 6. Comparison of illustrations between the BSB and CSB.** (A) Limitations when mapping the Medibloc service on the CSB. (B) Solutions to the limitations represented by mapping the Medibloc service on the BSB.

containing diverse components and complex service processes, was chosen as a representative example.

First, Medibloc was mapped on the CSB (Fig 6A), highlighting notable limitations of the CSB. The mapping revealed a lack of content in the Physical Evidence and Backstage Employee Actions rows. The Physical Evidence row presents physical things such as 'food' and 'car' as the word itself. Therefore, this row is limited in visualizing evidence such as tokens, coins, and transaction items that occur online (Limitation 1). This limitation makes it challenging for service developers to identify online evidence exchanged between users. Online evidence is frequently moved and is essential for blockchain services.

Second, the CSB can visualize only one user, and therefore cannot present actions of multiple users (Limitation 2). Blockchain was originally developed to stably implement a P2P system. Hence, interactions among multiple users are frequent. This problem causes confusion regarding which user interacts with the onstage employee on the operation platform. Therefore, service developers have difficulty understanding how users interact, and how the onstage employee responds to the requirements of users. Therefore, the CSB has limits on representing the unique characteristics of blockchain services.

Mapping Medibloc on the ISB (Fig 7A) reveals limitations similar to those of the CSB. Although the ISB can present transaction items, it cannot visualize tokens and coins (Limitation 1). Transaction items are physical and non-physical items exchanged during a service. Tokens and coins have different functions than other items exchanged within the service. Tokens and coins may represent specific physical items or information and may have a monetary value. Therefore, the Information row is insufficient for representing tokens and coins' unique functions. They are very important in the service process of blockchain services, so

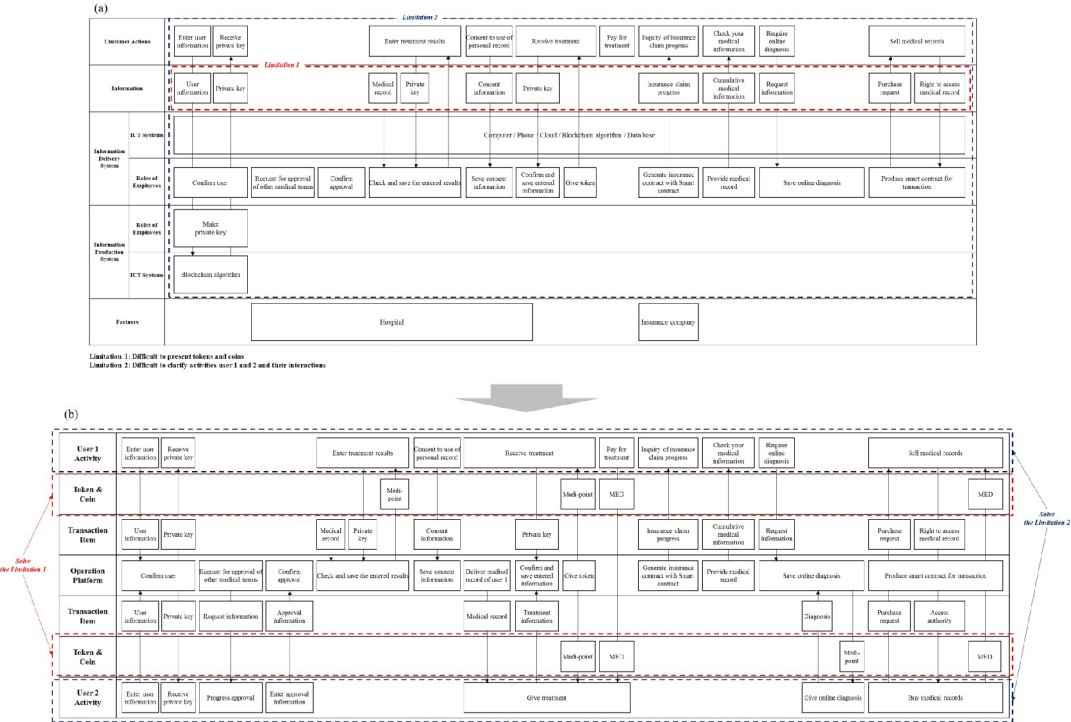

**Fig 7. Comparison of illustrations between the BSB and ISB.** (A) Limitations when mapping the Medibloc service on the ISB, (B) Solutions to the limitations represented by mapping the Medibloc service on the BSB.

because of this problem, service developers can have difficulty understanding the unique principles of blockchain services compared to existing services.

Finally, the ISB has limitations in presenting each process of multiple users (Limitation 2). The Information Product Systems row presents the actions of employees and ICT systems to produce information. Therefore, the row is not appropriate to present user activities. As such, the ISB is difficult to visualize activities and interactions of multiple users. Because of this limitation, service developers have difficulty understanding how each user interacts with the operation platform and how activities of users affect other users.

Mapping various blockchain services on the SEB (Fig 8A) showed similar limitations to those found in the CSB and ISB. The major problem is that the SEB has limitations in presenting multiple users in the services (Limitation 1). It similarly causes difficulty in understanding multiple users' interactions during the service. Additionally, the SEB emphasizes the actions of users, service providers, and support systems, so it is useful to understand their roles in a service. However, the SEB is limited in representing item exchanges between users and between a user and service provider. Therefore, the SEB complicates the task of identifying which transaction items, tokens, and items are exchanged during a service (Limitation 2).

The limitations Identified when mapping Medibloc on the CSB, ISB, and SEB can be solved by mapping on the BSB (Figs 6B, 7B and 8B). First, the limitations related to the process of tokens, coins, and transaction items can be solved using the Token & Coin rows and the Transaction Item rows. These rows enable visualization of service processes that use tokens, coins, and transaction items. This visualization with the BSB can help developers to understand which activities each user takes, and which token, coin, and transaction items are used during the service process. Additionally, the limitations in visualizing two-user interactions

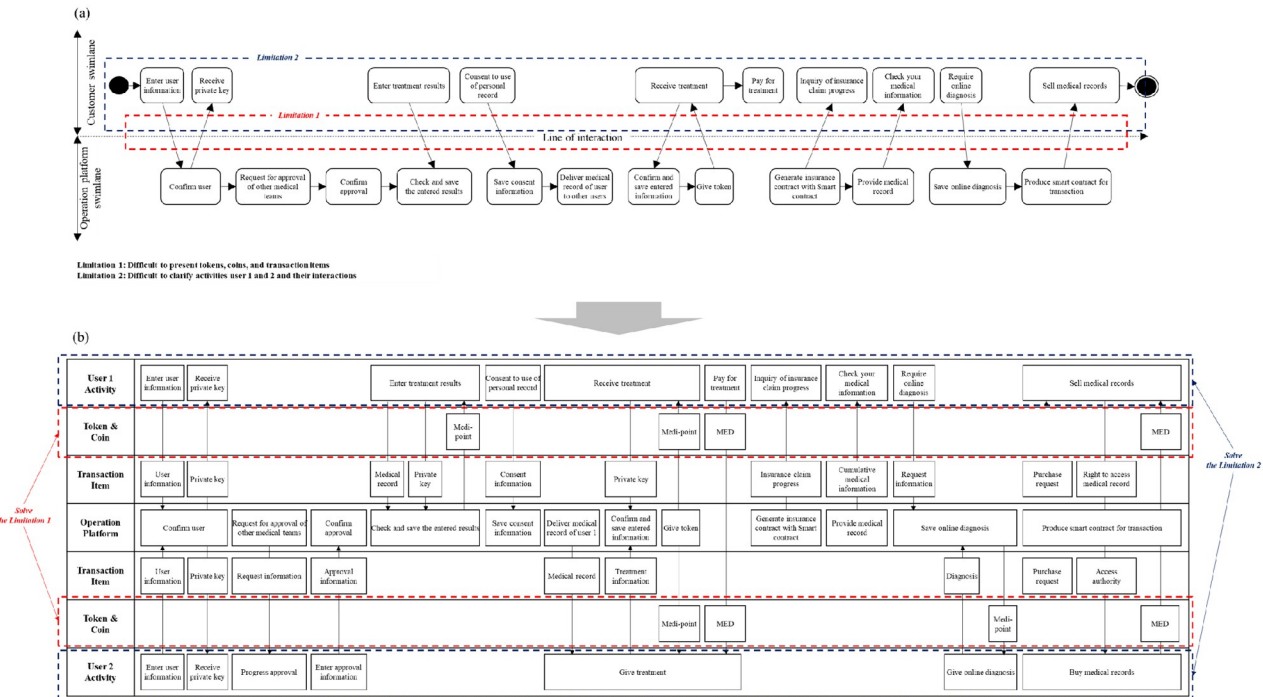

**Fig 8. Comparison of illustrations between the BSB and SEB.** (A) Limitations when mapping the Medibloc service on the SEB, (B) Solutions to the limitations represented by mapping the Medibloc service on the BSB.

are solved by the symmetric structure of the BSB. Especially, the BSB's user row can represent various stakeholder types and user groups. Users who use the SEB might draw multiple blueprints for each service provider and user, but the separate blueprints cannot easily present all service processes at a glance. The BSB presents all service processes where various service providers interact with different users in a single service delivery, so that users who use the BSB can reduce time and effort and can systematically identify important interactions that cannot be found with separate multiple blueprints.

## Managerial implications

This section discusses three managerial implications that are beneficial for users by utilizing the BSB. First, the proposed BSB enables service operators to identify problem points that may increase cost or reduce speed in processes in blockchain services. As blockchain services have new types of complex interactions with various participants in service processes [23], service developers or operators often find it challenging to identify unexpected or hidden problem points. The BSB illustrates various new interactions between users and a platform according to the sequence of users' activities. Such visualization provides a detailed process step by step and helps service providers and operators to review important points such as processes of private information and touch points with users. In Medibloc (Fig 4), a user in a medical team needs the approval of other users to register for the service. This process increases the security and the trust in a medical team but requires a long time for the user to participate in the service. This problem can be mitigated by sharing the information with government institutions that hold users' identification, such as medical license. In this example, the BSB helps service developers and operators to find potential problems and unnecessary processes. Such support can

prompt actions to resolve them, ultimately enhancing user experience. This is the BSB's advantage that includes key components for explaining processes of blockchain services, compared with existing SBs.

Second, the BSB can help to balance rewards for users' activities. One of the most important features in blockchain services is the reward system. In this system, setting the appropriate degree of reward is very important for the services. If a user can gain more rewards from simple activities (e.g., searching information and viewing content) than from important activities (e.g., suggesting changes to content and sharing important information), then the user conducts only simple activities. Additionally, if rewards are unlimitedly provided for every activity, then many meaningless activities can occur. As a result, improper reward systems decrease the quality of service. From this point of view, the BSB, which expresses the process of tokens and coins related to the user's activity in a separate intuitive row, can help establish a proper reward system. The goal of Everipedia (Fig 5) is to share high quality information. Therefore, the activities of proposing, changing and verifying contents is more important than providing personal information when signing up, to improve the quality of service. Thus, service quality can be effectively improved by allocating more rewards to content-related activities. Existing SBs may not be effective at taking this advantage, because they do not have appropriate components to visualize the process of tokens and coins. Such information can help service providers and operators to set a proper reward system.

Finally, the BSB can help identify opportunities to enhance blockchain services engagement. From the literature and service reviews, it is revealed that applying blockchain in service has changed the role and authority of users and brought a new type of interaction compared to existing similar services. However, it is not easy to induce the appropriate interaction between users because it requires a sufficient understanding of the service process. At this time, the BSB can be useful for activating interactions between users, which are important to maintain a blockchain service. For example, service providers can use the BSB to seize the appropriate opportunity where to introduce tokens and coins. Tokens and coins effectively facilitate user interaction, but indiscriminate use of them could reduce service value. In Medibloc (Fig 4), the activity that approves medical workers is important. Although approving other users is a time-consuming task, it is revealed that tokens or coins are not being paid in relation to the activity. Rewarding these activities can facilitate user verification. In this way, all areas where any tokens and coins are not presented, which are easily expressed in the BSB, can be potential areas to apply tokens and coins. Service developers can activate interactions between users by looking for a way to appropriately use tokens and coins in such areas. The proposed BSB and its implications are expected to inspire academic exploration, aiding in identifying business opportunities and enhancing customer experience.

## Concluding remarks

This study proposes the BSB, a novel tool developed to visualize and analyze the unique processes of blockchain services. The BSB was developed through an in-depth analysis of blockchain services' characteristics, followed by literature review and mapping of diverse service examples. The Medibloc and Everipedia services were used to validate the BSB's capacity to represent complex and varied blockchain processes. Furthermore, the BSB was evaluated through expert interviews, which confirmed its completeness, effectiveness, efficiency, and versatility as a visualization tool. A comparison with traditional blueprints, i.e., the CSB, ISB, and ESB, highlights the BSB's ability to overcome the limitations of these models, especially in

visualizing decentralized interactions and complex token and transaction flows that are specific to blockchain services.

The key findings from this study emphasize several implications. First, by integrating blockchain's technical properties, the BSB offers a more accurate representation of blockchain-specific service processes, which include user activities, token transactions, and operation platforms. This helps clearly communicate the architecture of blockchain services. Second, the case illustrations of Medibloc and Everipedia demonstrate the BSB's flexibility in mapping various blockchain services, effectively capturing both complex and decentralized interactions. Finally, expert interviews confirmed the BSB's practical value, particularly in improving communication within cross-functional teams and supporting service developers in optimizing processes. The BSB's generality and versatility make it a useful tool for applying to a wide range of blockchain services, addressing challenges that traditional service blueprints are not equipped to handle.

This study has the following contributions. First, by considering the technical characteristics of blockchain and its application to services, this research accurately represented the changes blockchain brings to the service process. The literature and blockchain services were reviewed, and meaningful insights were extracted as key components. Second, the development of a service blueprint specialized for blockchain services was based on a deep understanding of blockchain services. The BSB provides a comprehensive view of the entire system, such as users, platform, tokens, and coins, at a glance. This visualization tool is expected to help users in various purposes. Finally, the managerial implications were discussed for service developers and designers. These implications help them take full advantages of the BSB for their work.

The BSB presents several promising applications but also has limitations that future research could address. First, one limitation is the reliance on manual data input and interpretation, which may lead to inconsistencies and inefficiencies in mapping blockchain service processes. Automating the BSB through software development could address this issue. Such software would collect and analyze data from decentralized applications (DApps) across different sectors, assigning data to relevant sections of the BSB. By incorporating customer behavior and event log data, automated tools could improve the BSB's accuracy and usability, offering practitioners real-time insights and reducing the time needed to create comprehensive service maps.

Another limitation of the current BSB design is its generalized structure, which may not fully accommodate the diver types of blockchain networks, such as public, private, and consortium blockchains. Each type has unique characteristics; for instance, public blockchains allow open access, while private and consortium blockchains restrict access based on permissions set by service operators. These distinctions impact service processes and user interactions. Future research could focus on developing specialized BSB variations tailored to these specific blockchain types. Modifying the BSB to reflect the permissions, governance structures, and data accessibility of each blockchain type would enable more accurate and nuanced visualization of service processes, enhancing the blueprint's practical application for service developers and providers.

Lastly, there is a need to further examine the integration of machine learning and predictive analytics within the BSB framework. By leveraging data-driven insights, the BSB could evolve to not only map current processes but also predict future service needs and improvements based on user behavior and network activity. Research in this direction could empower the SBS to move from a descriptive to a predictive tool, providing valuable foresight for blockchain service providers and enabling more proactive service management.

## Author Contributions

**Conceptualization:** Do-Hyeon Ryu.

**Data curation:** Hyeji Jang.

**Formal analysis:** Do-Hyeon Ryu.

**Supervision:** Do-Hyeon Ryu.

**Validation:** Hyeji Jang.

**Writing – original draft:** Hyeji Jang, Do-Hyeon Ryu.

**Writing – review & editing:** Hyeji Jang, Do-Hyeon Ryu.

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
