## [Decision Letter · Decision Letter 0]

4 Oct 2024

PONE-D-24-25304Development of a Service Blueprint for Blockchain ServicesPLOS ONE

Dear Dr. Ryu,

Thank you for submitting your manuscript to PLOS ONE. After careful consideration, we feel that it has merit but does not fully meet PLOS ONE’s publication criteria as it currently stands. Therefore, we invite you to submit a revised version of the manuscript that addresses the points raised during the review process.

We look forward to receiving your revised manuscript.

Kind regards,

Barbara Guidi

Academic Editor

PLOS ONE

Journal Requirements:

NO authors have competing interests. 

6. Please amend your list of authors on the manuscript to ensure that each author is linked to an affiliation. Authors’ affiliations should reflect the institution where the work was done (if authors moved subsequently, you can also list the new affiliation stating “current affiliation:….” as necessary). 

Reviewers' comments:

Reviewer's Responses to Questions

**Comments to the Author**

1. Is the manuscript technically sound, and do the data support the conclusions?

Reviewer #1: Yes

Reviewer #2: Yes

2. Has the statistical analysis been performed appropriately and rigorously? 

Reviewer #1: Yes

Reviewer #2: Yes

3. Have the authors made all data underlying the findings in their manuscript fully available?

Reviewer #1: Yes

Reviewer #2: Yes

4. Is the manuscript presented in an intelligible fashion and written in standard English?

Reviewer #1: Yes

Reviewer #2: Yes

5. Review Comments to the Author

Reviewer #1: 1- Choosing keywords that are different from the words in the title

2- Ensure that research papers are included in related works according to the year of publication (from oldest to newest)

3- References are arranged in Table 1 according to publication date, from oldest to newest.

4- Figures 4-8 need to be reconstructed due to unclear texts.

5- Conclusions need to be rephrased and defined based on the experiments the researcher has conducted.

6- Additional references published in 2024 are included.

Reviewer #2: Strengths:

1. I really loved the paper's key innovation lies in its development of a blueprint specifically tailored for blockchain services. It extending traditional service blueprints by including blockchain-specific elements such as decentralized user interactions, smart contracts, and token/coin transactions.

2. This framework is very useful for service providers aiming to improve their processes and enhance service sustainability, offering broad applications across industries such as healthcare, financial services, and social networks.

3. The paper is well-structured, with clear delineations between the literature review, methodology, and results. The introduction effectively highlights the need for the BSB, and the illustrations provided (e.g., Medibloc and Everipedia) clearly demonstrate the utility of the blueprint.

Recommendations:

However, there are areas where the explanation could be condensed, particularly in the literature review section, to focus more on the blueprint's practical applications. Additionally, further elaboration on the potential limitations of the BSB and future areas of research, such as the development of automated BSB tools, would strengthen the paper.

6. PLOS authors have the option to publish the peer review history of their article (what does this mean?). If published, this will include your full peer review and any attached files.

Reviewer #1: No

Reviewer #2: **Yes: **Mounica Achanta

---

## [Author Response · Author response to Decision Letter 0]

12 Nov 2024

Development of a Service Blueprint for Blockchain Services

We would like to thank the editor and reviewers for helpful and insightful comments and suggestions that helped significantly improve this manuscript. The comments are addressed below.

Response to Editor’s Comments

Comment 1.

Response: We have revised the manuscript in accordance with PLOS ONE’s style requirements based on the provided samples for guidance, including the file naming.

Comment 2.

Please note that PLOS ONE has specific guidelines on code sharing for submissions in which author-generated code underpins the findings in the manuscript. In these cases, all author-generated code must be made available without restrictions upon publication of the work.

Response: No code was used in this study.

Comment 3.

Please complete your Competing Interests on the online submission form to state any Competing Interests. If you have no competing interests, please state ""The authors have declared that no competing interests exist."", as detailed online in our guide for authors at http://journals.plos.org/plosone/s/submit-now This information should be included in your cover letter; we will change the online submission form on your behalf.

Response: In accordance with the online submission format, we have completed the Competing Interests section in the cover letter by stating, “The authors have declared that no competing interests exist.”

Comment 4.

Please provide a complete Data Availability Statement in the submission form, ensuring you include all necessary access information or a reason for why you are unable to make your data freely accessible. If your research concerns only data provided within your submission, please write "All data are in the manuscript and/or supporting information files" as your Data Availability Statement.

Response: As this study utilized literature review, qualitative analysis (e.g., open card sorting, and comparison with existing service blueprints), and expert discussion and evaluation, no data was collected. Therefore, we have confirmed that there is no data to share.

Comment 5.

PLOS requires an ORCID iD for the corresponding author in Editorial Manager on papers submitted after December 6th, 2016. Please ensure that you have an ORCID iD and that it is validated in Editorial Manager. To do this, go to ‘Update my Information’ (in the upper left-hand corner of the main menu), and click on the Fetch/Validate link next to the ORCID field. This will take you to the ORCID site and allow you to create a new iD or authenticate a pre-existing iD in Editorial Manager.

Response: The corresponding author’s ORCID iD has been registered and validated following the procedure provided.

Comment 6.

Please amend your list of authors on the manuscript to ensure that each author is linked to an affiliation. Authors’ affiliations should reflect the institution where the work was done (if authors moved subsequently, you can also list the new affiliation stating “current affiliation:….” as necessary).

Response: We have reviewed and revised the affiliations of all authors to ensure that each author is accurately linked to their respective institution.

Comment 7.

Response: We have reviewed all references and confirmed that there are no issues with the cited references.

Response to Reviewer 1’s Comments 

Comment 1.1.

Choosing keywords that are different from the words in the title.

Response: We have reviewed the keywords in accordance with your comments, reconsidered overlapping terms, and added a few that are not included in the title. As a result, we have finalized the keywords as ‘service blueprint, service process design, blockchain services, service delivery visualization.’

Comment 1.2.

Ensure that research papers are included in related works according to the year of publication (from oldest to newest).

Response: Additional research papers related to blockchain and blockchain services were reviewed, with a particular focus on publications from 2020 to 2024. This effort allowed us to update references with the latest studies and enhance the content accordingly.

Comment 1.3.

References are arranged in Table 1 according to publication date, from oldest to newest.

Response: In response to Comment 1.2, we have updated the manuscript with the latest references. Additionally, the references have been reorganized in sequential order following the reference style guidelines of PLoS ONE.

Comment 1.4.

Figure 4-8 need to be reconstructed due to unclear texts.

Response: The original Figure 4-8 were adjusted to increase text size in proportion to the blueprint size. Additionally, high-resolution settings were applied to ensure clarity when adding the figures to the manuscript.

Comment 1.5.

Conclusions need to be rephrased and defined based on the experiments the researcher has conducted.

Response: We have restructured the content of the concluding remarks appropriately based on the experiments conducted in this study. The revised concluding remarks now clearly reflect the experiments performed and the significance of their outcomes.

Comment 1.6.

Additional references published in 2024 are included.

Response: As stated in the response to Comment 1.2, additional 2024 publications have been reviewed and incorporated to further strengthen the references.

Response to Reviewer 2’s Comments 

Comment 2.1.

There are areas where the explanation could be condensed, particularly in the literature review section, to focus more on the blueprint's practical applications.

Response: As per your feedback, certain sections were indeed lengthy and overly detailed. A comprehensive review and revision have been conducted across the paper to identify areas for condensation within each paragraph. Adjustments were made to enhance the focus on the practical applications of blockchain services and blueprints, as well as to underscore the value of the BSB.

Comment 2.2.

Additionally, further elaboration on the potential limitations of the BSB and future areas of research, such as the development of automated BSB tools, would strengthen the paper.

Response: As per your comment, we recognized that a detailed discussion of the BSB’s limitations and potential areas for future research would be valuable for readers. In response, we expanded the concluding remarks section by identifying three key limitations of the BSB (in three separate paragraphs) and proposing corresponding solutions for each limitation. This enhancement aims to provide a more comprehensive overview of future directions for addressing these challenges.

---

## [Decision Letter · Decision Letter 1]

30 Dec 2024

Development of a service blueprint for blockchain services

PONE-D-24-25304R1

Dear Dr. Ryu,

We’re pleased to inform you that your manuscript has been judged scientifically suitable for publication and will be formally accepted for publication once it meets all outstanding technical requirements.

Kind regards,

Barbara Guidi

Academic Editor

PLOS ONE

Additional Editor Comments (optional):

Reviewers' comments:

Reviewer's Responses to Questions

**Comments to the Author**

1. If the authors have adequately addressed your comments raised in a previous round of review and you feel that this manuscript is now acceptable for publication, you may indicate that here to bypass the “Comments to the Author” section, enter your conflict of interest statement in the “Confidential to Editor” section, and submit your "Accept" recommendation.

Reviewer #1: All comments have been addressed

Reviewer #2: All comments have been addressed

2. Is the manuscript technically sound, and do the data support the conclusions?

Reviewer #1: (No Response)

Reviewer #2: Yes

3. Has the statistical analysis been performed appropriately and rigorously? 

Reviewer #1: (No Response)

Reviewer #2: Yes

4. Have the authors made all data underlying the findings in their manuscript fully available?

Reviewer #1: (No Response)

Reviewer #2: Yes

5. Is the manuscript presented in an intelligible fashion and written in standard English?

Reviewer #1: (No Response)

Reviewer #2: Yes

6. Review Comments to the Author

Reviewer #1: (No Response)

Reviewer #2: All the review comments has been addressed. Thank you for providing more clarity and depth in the concept.

7. PLOS authors have the option to publish the peer review history of their article (what does this mean?). If published, this will include your full peer review and any attached files.

Reviewer #1: No

Reviewer #2: **Yes: **Mounica Achanta

---

## [Editor Report · Acceptance letter]

16 Jan 2025

PONE-D-24-25304R1 

PLOS ONE

Dear Dr. Ryu, 

I'm pleased to inform you that your manuscript has been deemed suitable for publication in PLOS ONE. Congratulations! Your manuscript is now being handed over to our production team.

Kind regards, 

on behalf of

Dr. Barbara Guidi 

Academic Editor

PLOS ONE